# Pharmacological Treatments against COVID-19 in Pregnant Women

**DOI:** 10.3390/jcm10214896

**Published:** 2021-10-24

**Authors:** Ana Arco-Torres, Jonathan Cortés-Martín, María Isabel Tovar-Gálvez, María Montiel-Troya, Blanca Riquelme-Gallego, Raquel Rodríguez-Blanque

**Affiliations:** 1Facultad de Ciencias de la Salud, Universidad de Granada, Grupo de Investigación CTS-1068, 18014 Granada, Spain; b1708@correo.ugr.es (A.A.-T.); jonathan.cortes.martin@gmail.com (J.C.-M.); 2Departamento de Enfermería, Facultad de Ciencias de la Salud, Universidad de Granada, Grupo de Investigación CTS-1068, 18014 Granada, Spain; mariamontiel@ugr.es (M.M.-T.); briquel@ugr.es (B.R.-G.); 3Delegación Territorial de Salud y Familias, Grupo de Investigación CTS-1068, 18014 Granada, Spain; raquel.rodriguez.blanque.sspa@juntadeandalucia.es

**Keywords:** SARS-CoV-2, COVID-19, virus, pregnant woman, foetus, pharmacological treatment

## Abstract

The recent respiratory virus known as SARS-CoV-2 has caused millions of deaths worldwide, causing great uncertainty due to the lack of a specific treatment, which has been mitigated by the use of various drugs traditionally used against other types of pathologies. Pregnancy presents special physiological conditions that expose the pregnant woman and the foetus to greater risk. Pregnant women are often excluded from trials due to possible risk of toxicity or side effects, resulting in a lack of knowledge about the use of drugs and treatments during pregnancy. The main objective of this review was to compile existing knowledge about currently available drug treatments for COVID-19 in pregnant women. The review report met the criteria of the Preferred Reporting Items for Systematic reviews and Meta-Analyses (PRISMA) review protocol and was registered with the registration number CRD42021251036. The electronic databases searched were Scopus, PubMed, CINAHL and SciELO. Finally, 22 articles were included, resulting in an analysis of drugs with an acceptable safety profile in the treatment of pregnant women with COVID-19.

## 1. Introduction

Several episodes of pneumonia of unknown origin were reported in December 2019 in Wuhan province, China, and these were assigned to a new coronavirus linked to the 2003 severe acute respiratory syndrome-CoV (SARS-CoV), and thus designated SARS 2 (SARS-CoV-2) [1,2,3].

By early 2020, SARS-CoV-2 had caused more than 85 million cases of coronavirus disease 2019 (COVID-19) and about 2 million deaths worldwide [4]. The rapid spread of the virus led the World Health Organization (WHO) to proclaim this outbreak a pandemic on 11 March 2020 [2,3,5].

SARS-CoV-2 is highly infectious and is transmitted by respiratory droplets, which are now known to have a higher magnitude of infectivity than previously assumed [5]. SARS-CoV-2 infection can present as asymptomatic, mild or severe, even leading to compromised respiratory failure [2].

During pregnancy, important physiological changes occur (significant increase in volume of distribution, lower albumin concentrations, hormone-induced susceptibility to liver toxicities, etc.) [6], leading to a state of partial immunosuppression, which results in increased vulnerability to viral infections [4,5,6,7,8,9].

Angiotensin-converting enzyme (ACE)-2, which is the receptor for SARS-CoV-2, is significantly increased in the process of pregnancy, with the consequence that pregnant women are more susceptible to SARS-CoV-2 [10]. Consequently, the COVID-19 pandemic may have dire consequences for pregnant women [8,11].

Drugs used during pregnancy and lactation have been categorised by the Food and Drug Administration (FDA) into five groups (A, B, C, D and X), distinguishing these according to the risk of teratogenicity and maternal, foetal and neonatal complications. Category A introduces drugs that have not demonstrated a risk to the foetus in the first trimester of pregnancy (and for which there is no evidence of risk in later trimesters). Category B are drugs for which no risk to the foetus has been shown in animal reproduction studies, but for which there are no adequate and well-controlled studies in pregnant women. Category C are drugs for which animal reproduction studies indicate an adverse effect on the foetus and for which adequate and well-controlled studies in humans are lacking. However, the potential benefits may argue for the use of the drug in pregnant women despite the potential risks. Category D pertains to drugs that have certain evidence of human foetal risk based on adverse reaction data; however, despite the potential risks, the benefits may justify the use of the drug in this population. Category X includes drugs for which animal or human studies have shown foetal abnormalities and/or positive evidence of human foetal risk [2].

The importance of this literature review is based on the current scarcity of studies offered for the treatment of COVID-19 in pregnant women, as well as the need to homogenise concepts to extrapolate valid results that can be put into clinical practice, thus ensuring safe and optimal treatments for the mother and foetus [12].

## 2. Materials and Methods

The methodology used for the elaboration of this report was a literature review of the published scientific literature on pharmacological treatments in pregnant women with COVID-19, following the Preferred Reporting Items for Systematic reviews and Meta-Analyses (PRISMA) review protocol, which consists of a 27-point checklist of the most representative sections of an original article, as well as the process of elaboration of these guidelines. This systematic review has been carried out following a protocol, available on the web at http://www.crd.york.ac.uk/PROSPERO/ (accessed date 05/2021) and whose registration number is CRD42021251036. We selected current articles up to five years old, providing information on pharmacological treatments in pregnant women with COVID-19, with no restrictions on language of publication or type of article. The literature search was conducted in the Scopus, PubMed, CINAHL and SciELO databases, and a manual search was performed using reference lists of studies to find other relevant studies. The structured language used was that obtained using Medical Subject Headings (MeSH) terms and Health Sciences (DeCS) descriptors. The descriptors used were “SARS-CoV-2”, “COVID19”, “pharmacotherapy” and “pregnancy”, and the Boolean operators used were AND and OR.

All articles found were transferred to the Mendeley web application using the Mendeley Web Importer tool. After exporting all the articles to the Mendeley website, they were organised by folders according to the database from which they had been obtained. Finally, all duplicates were eliminated, leaving a definitive study list. Once the list was created, a selective reading of the titles and abstracts of all the studies found was carried out to select those that could be related to the data to be collected. Then, an individualised reading of each article was carried out to check whether it met the inclusion criteria, discarding the documents in which no relationship was found with the objectives and characteristics of this review.

To carry out the methodological evaluation of the 22 articles selected for this study, the design, methodology and type of study of each paper were analysed to select the most specific methodological evaluation scale for each case. Of the 22 articles, 20 were literature reviews and 2 were case studies, which provide a detailed account of the symptoms, medical signs, diagnoses and therapeutic follow-up of pregnant women diagnosed with COVID-19, the main focus of this paper.

Articles with a case-study design were assessed using the Single-Case Experimental Design (SCED) Scale. The SCED was constructed including 11 items, of which 10 were used to assess methodological quality and 1 for the use of statistical analysis [13].

The methodological assessment scale Amstar-2 (A MeaSurement Tool to Assess systematic Reviews) was used for the reviews. Amstar-2 provides a broad assessment of quality, incorporating imperfections that may have arisen due to improper conduct of the review. Amstar-2 was constructed to include 16 domains, which present simple response options: “yes” when the product is positive, “no” if the standard was not met or the existing information was too sparse to answer and “partial yes” in situations where partial adherence to the standard was given. Although it does not provide an overall rating, four levels of confidence emerge: high, moderate, low and critically low [13,14].

## 3. Results

Based on the information provided by this review, a series of premises were obtained that served to homogenise concepts on the optimal drugs for the treatment of pregnant women with COVID-19. In addition to an analysis of pharmacological therapy, the existence of few studies offered to pregnant women was confirmed.

The results of the selected articles are shown in Table 1. Articles were grouped according to the different groups of drugs used in the treatment of SARS-CoV-2 in pregnant women: anticoagulants, corticosteroids, convalescent plasma, nitric oxide and antiviral drugs.

There is now increasing evidence of a significant risk of thromboembolic events in patients with COVID-19, as stated by Favilli et al. [12]. Authors, such as Cavalcante et al. [2], Favilli et al. [12] and Lat et al. [15], advise immediate initiation of prophylaxis in all pregnant women with COVID-19, as pregnancy itself is known to be a thrombotic condition. According to Favilli et al. [12] and Lat et al. [15], low-molecular-weight heparin should be administered in women who are not close to the time to delivery.

Recent evidence supports the use of an early and short course of glucocorticoids in pregnant patients with COVID-19 who require mechanical ventilation or oxygen support, as verified by authors Saad et al. [16] and Souza et al. [4]. Saad et al. [16], Louchet et al. [6] and Mcintosh [17] state that, of all the corticosteroids, the two optimal ones for inducing foetal lung maturity, despite increasing blood glucose levels, are betamethasone and dexamethasone, as they have the highest rate of placental transfer with minimal mineralocorticoid effects. Saad et al. [16] state that exposure to periodic cycles of prenatal glucocorticoids has been associated with adverse neurological outcomes, small head circumference, foetal growth restriction and increased risk of neonatal hypoglycaemia.

There is evidence that SARS-CoV-2 infection throughout pregnancy may increase the risk of premature rupture of membranes, impairing foetal development and preterm delivery, and therefore betamethasone administration is justified, as verified by Favilli et al. [12]. Finally, Souza et al. [4] express the importance of carefully weighing the risks and benefits of using corticosteroids in this type of patient and that administration should not delay urgent delivery.

On the use of convalescent or hyperimmune plasma in pregnant women with COVID-19, there are no well-controlled studies on its use, as stated by authors Cavalcante et al. [2], Favilli et al. [12] and Lat et al. [15]. Cavalcante et al. [2] suggest that hyperimmune plasma may be of greater therapeutic value if administered during the early course of the disease, i.e., within 14 days of symptom onset. Based on the results described previously, hyperimmune plasma does not prove to be an effective treatment for COVID-19 in pregnant women at the moment.

Safaee et al. [18] state that nitric oxide at high doses (160–200 ppm) has shown high antimicrobial activity on bacteria and viruses such as SARS-CoV-1, which could be favourable in pregnant patients with COVID-19 manifesting hypoxic respiratory failure, but this needs to be demonstrated in prospective randomised trials. A study of a small number of pregnant women with COVID-19 demonstrates improved cardiopulmonary function after starting nitric oxide gas [18].

With regard to antiviral drugs, a selection has been made by choosing those with the most evidence and the largest number of studies supporting the results.

Remdesivir has shown in vitro activity against several new types of coronavirus (nCoV), SARS-CoV-2 among them, so the use of this drug could represent a therapy for COVID-19 thanks to its broad range, as mentioned by Favilli et al. [12], Liang et al. [8] and Castro et al. [19]. However, authors such as Bardon et al. [20], Cavalcante et al. [2], Lat et al. [15], Louchet et al. [6] and Taylor et al. [21], state that the use of remdesivir in pregnant women has not been exhaustively studied, although animal studies did not reveal adverse effects on embryonic development.

Cavalcante et al. [2] advocate its use only if the potential benefit justifies the potential risk to the mother and foetus. Louchet et al. [6] argue against its use in pregnant women because of its hepatic and renal toxicity profile; given its small weight and high protein-binding rate, remdesivir can be expected to cross the placenta.

Cavalcante et al. [2] and FaviIli et al. [12] argue that there is a paucity of data available on the use of chloroquine (CQ) or hydroxychloroquine (HCQ) in the treatment of pregnant women with COVID-19. Currently the existing data on the safety of these drugs in pregnant women comes from studies on the treatment of malaria. Regarding the toxicity caused by these drugs, Cavalcante et al. [2], Louchet et al. [6], Giampreti et al. [22] and Li et al. [23] report possible ototoxic and retinotoxic risks, as chloroquine and hydroxychloroquine cross the placenta by passive diffusion. Louchet et al. [6] report that the risk of miscarriage has not been increased with the use of these drugs. Although both drugs share similar therapeutic effects, Zhao et al. [24] report higher side effects for CQ and lower side effects for HCQ. With respect to the treatment schedule, according to Favilli et al. [12], CQ can be used in doses of 1 g on the first day of treatment, then 500 mg per day for 4–7 days. Treatment with HCQ is based on oral administration of or 400 mg every 12 h for 5 days, or 400 mg twice daily for the first day and then 200 mg twice daily for 4 days. The patient’s clinical response to treatment needs to be evaluated. Daily maternal doses of less than 400 mg have not been associated with adverse maternal outcomes, which is corroborated by Lat et al. [15]. Li et al. [23] warn of the importance of evaluating the advantages and disadvantages before use in pregnant women with COVID-19. Note that HCQ is a treatment also used in patients with autoimmune diseases [25].

As for pregnancy, Favili et al. [12] and Giampreti et al. [22] reason that no data are available on the effects of lopinavir (LPV)/ritonavir (r) in the treatment of pregnant women with COVID-19. 

Authors such as Cavalcante et al. [2], Favilli et al. [12] and Louchet et al. [6] discuss the use of these drugs in pregnant women for the treatment of the HIV virus, thus providing evidence and safety of these drugs during pregnancy. However, results such as reported by Cavalcante et al. [2], Bardon et al. [20] and Giampreti et al. [22] indicate that administration of LPV/r to pregnant women presents a similar prevalence of fetal malformations and obstetric difficulties to pregnant women who did not use them. Lambelet et al. [3] articulated the possibility of moderate adverse events, such as an increased risk of impaired fasting blood glucose and gastrointestinal symptoms. If possible, oral solution should be avoided throughout pregnancy due to the alcohol and propylene glycol content, as stated by Giampreti et al. [22] and Li et al. [23]. Zhao et al. [24] and Cavalcante et al. [2] argue that LPV/r is the preferred therapy for pregnant women with COVID-19.

Giampreti et al. [22] assert that experimental animal studies and human reports show no expectation of increased risk during pregnancy from the use of azithromycin; therefore, Giampreti et al. [22], Cavalcante et al. [2] and Louchet et al. [6] affirm the compatibility and safety of azithromycin use in the treatment of COVID-19 during pregnancy.

Cavalcante et al. [2] state that azithromycin given at a dose two to four times higher than the dose administered to humans was not associated with any evidence of foetal harm. Observational studies in humans have not revealed a major risk of teratogenic or obstetric complications associated with azithromycin use during pregnancy. In terms of the effects of the use of this drug, authors, such as Cavalcante et al. [2] and Favilli et al. [12], report that the most common are diarrhoea, vomiting, nausea, abdominal pain and, less frequently, a change in the electrical activity of the heart, specifically by increasing the QT interval.

**Table 1 jcm-10-04896-t001:** Most relevant studies in this review.

AUTHOR	YEAR	ARTICLE	TYPE OF ARTICLE	OBJECTIVES	RESULTS
Favilli et al. [12]	2020	Effectiveness and safety of available treatments for COVID-19 during pregnancy: a critical review.	Critical review	To provide a review of the literature on the presumed effectiveness and safety of available treatments for COVID-19 in pregnant women.	Pregnant women represent a fragile category of patients, generally excluded from trials, and the choice to use a COVID-19-specific drug must take into account the benefits and potential adverse events in each case.
Malhamé et al. [11]	2020	The Moral Imperative to Include Pregnant Women in Clinical Trials of Interventions for COVID-19.	Literature review	Argue for the importance of including pregnant people in clinical trials of interventions for COVID-19.	The exclusion of pregnant women from clinical trials reduces the external validity of the study results.
Louchet et al. [6]	2020	Placental transfer and safety in pregnancy of medications under investigation to treat coronavirus disease 2019.	Literature review	Analyse a number of drugs that may be compatible with pregnant women who have COVID-19.	Several drugs can now be used to treat pregnant women, but more pre-clinical studies are needed.
Safaee et al. [18]	2020	High Concentrations of Nitric Oxide Inhalation Therapy in Pregnant Patients With Severe Coronavirus Disease 2019 (COVID-19).	Case studies	To test the efficacy of nitric oxide in pregnant patients with COVID-19 and respiratory failure.	Nitric oxide at 160–200 ppm is easy to use and well tolerated and therefore could be beneficial in pregnant COVID-19 patients with hypoxic respiratory failure.
Liang et al. [8]	2020	Novel coronavirus disease (COVID-19) in pregnancy: What clinical recommendations to follow?	Literature review	To provide appropriate clinical management and support to pregnant patients, adopting a multidisciplinary team approach.	Pregnancy is a state of partial immunosuppression that makes pregnant women more vulnerable to viral infections.
Cavalcante et al. [2]	2021	COVID-19 Treatment: Drug Safety Prior to Conception and during Pregnancy and Breastfeeding.	Literature review	To assess the state of the current literature on the safety of therapies prescribed to treat COVID-19 during pregnancy.	There is currently no specific effective treatment for COVID-19, but a large number of drugs are being used to combat SARS-CoV-2 infection.
Taylor et al. [21]	2021	Inclusion of pregnant women in COVID-19 treatment trials: a review and global call to action.	Literature review	Provide pregnant women with information on the potential benefits and risks of exposure to candidate drugs for COVID-19 treatment.	The inclusion of pregnant women in clinical treatment trials is urgently needed to identify an effective COVID-19 treatment for this population.
Lat et al. [15]	2020	Therapeutic options in the treatment of severe acute respiratory syndrome coronavirus 2 in pregnant patients.	Literature review	To provide a review of available treatments for COVID-19 in pregnant women.	There are few case studies of SARS-CoV-2 therapies in pregnant patients, so the initiation of any drug therapy should be a comprehensive, multidisciplinary risk-benefit discussion.
Giampreti et al. [22]	2020	Medications prescriptions in COVID-19 pregnant and lactating women: The Bergamo Teratology Information Service experience during COVID-19 outbreak in Italy.	Case studies	To evaluate the efficacy and safety in pregnant women of various drugs for SARS-CoV-2 infection.	It should be noted that the safety of medicines during pregnancy is a constantly changing issue.
Bardon et al. [20]	2020	How should we treat pregnant women infected with SARS-CoV-2?	Literature review	Raise public awareness of the importance of not excluding pregnant women with COVID-19 from studies.	Pregnant women are as exposed as the general population and should not be excluded from discussions on effective and well-tolerated candidate treatments because of their status.
Saad et al. [16]	2020	Corticosteroids in the Management of Pregnant Patients With Coronavirus Disease (COVID-19).	Literature review	Assess possible initiation of maternal corticosteroid therapy and possible side effects.	Pregnant women with COVID-19 who require oxygen therapy or mechanical ventilation, or both, should be considered for steroid therapy.
Lai et al. [10]	2020	Severe acute respiratory syndrome coronavirus-2 and the deduction effect of angiotensin-convertingenzyme 2 in pregnancy.	Literature review	To analyse a case series of pregnant women with COVID-19.	SARS-CoV-2 is an ongoing disease, and the experience of pregnant women with SARS-CoV-2 is limited.
Mcintosh [17]	2020	Corticosteroid Guidance for Pregnancy during COVID-19 Pandemic.	Literature review	To examine the risks and benefits of corticosteroid therapy in pregnant women with COVID-19.	There is limited evidence on corticosteroid treatment in pregnant women with COVID-19. Therefore, a careful assessment of maternal risk versus neonatal benefit should be made.
Thomas et al. [7]	2020	Maternal and perinatal outcomes and pharmacological management of COVID-19 infection in pregnancy: A systematic review protocol.	Systematic review	To summarise the existing data on the effects of COVID-19 in the maternal population, address the therapeutic management and safety of drugs to treat COVID-19 during pregnancy and lactation.	There is a paucity of information related to pharmacological management and maternal and perinatal outcomes during the pandemic.
D’Souza et al. [4]	2021	Pregnancy and COVID-19: pharmacologic considerations.	Literature review	To summarise the evidence regarding the use of routine and investigational pharmacological interventions for pregnant patients with COVID-19.	The vast majority of common antepartum pharmacological interventions can be used in women with COVID-19.
Li et al. [23]	2020	Antiviral agent therapy optimization in special populations of COVID-19 patients.	Literature review	Review articles that focus on antiviral agents in patients with COVID-19 and summarise possible antiviral options against COVID-19.	Individual antiviral treatment should be carried out to achieve better clinical outcomes and to avoid adverse drug reactions.
Lambelet et al. [3]	2020	SARS-CoV-2 in the context of past coronaviruses epidemics: Consideration for prenatal care.	Literature review	To describe the current knowledge on the risks and consequences of COVID-19 in pregnancies, to summarise the current treatment options available to pregnant women, and finally, to compare the current guidance proposed by RCOG, ACOG and WHO to provide an overview of the prenatal management that should be used until future data become available.	More comprehensive data are needed to understand the additional risks that pregnancy may pose to women with COVID-19 infection.
Whitehead et al. [9]	2020	Consider pregnancy in COVID-19 therapeutic drug and vaccine trials.	Literature review	To analyse a case series of pregnant women with COVID-19.	The COVID-19 pandemic highlights the vulnerability of sick pregnant women if they are systematically excluded from clinical trials.
Zhao et al. [24]	2020	Analysis of the susceptibility to COVID-19 in pregnancy and recommendations on potential drug screening.	Bibliographic review	To analyse the susceptibility of SARS-CoV-2 in pregnancy and the drugs that can be used to treat pregnancy with COVID-19 and to provide evidence for drug selection in the clinic.	Large-scale investment is needed in research and development of vaccines and therapeutic drugs against coronavirus, in which pregnant women are a group that cannot be excluded.
Castro et al. [19]	2020	COVID-19 and Pregnancy: An Overview.	Literature review	Provide a brief discussion of COVID-19, pregnancy in the COVID-19 era and the effects of COVID-19 on pregnancy.	With the onset of the pandemic, research should be conducted on the effects of viraemia during the first and second trimester and the prediction of possible adverse outcomes.
Afshar et al. [5]	2020	Clinical guidance and perinatal care in the era of coronavirus disease 2019 (COVID-19).	Literature review	To review a case series of pregnant women with COVID-19.	At this time, there remain significant gaps in the evidence to enable comprehensive counselling of pregnant women and their families, specifically with respect to outcome risks.
Peyronnet et al. [1]	2020	Infection with SARS-CoV-2 in pregnancy. Update of Information and proposed care. CNGOF.	Literature review	To analyse a case series of pregnant women with COVID-19.	There is a need to continue to collect data on clinical cases of SARS-CoV-2 infection during pregnancy and to improve our understanding of the course of the disease during pregnancy.

## 4. Discussion

Historically, the exclusion of pregnant women from trials has led to a great deal of misinformation about the use of therapeutic agents during pregnancy [15], causing important consequences, such as refusal or delay of treatment due to concerns about exposure to a potentially harmful agent, the potential risk of improper treatment due to the use of unreliable and ineffective therapies throughout pregnancy, the likelihood of under- or overtreatment and incorrect dosing based on lack of pregnancy-specific pharmacokinetic data, potentially leading to unanticipated foetal and/or maternal toxicity [11]. The lack of evidence reinforces the need for non-exclusion of pregnant women from trials, in approaching the requirement for efficient treatment and the needs to have the clinical course of these women, with and without treatment, monitored and documented [21]. There are currently a number of drugs used in pregnant women with COVID-19 during pregnancy, some of which may pose toxicity problems to the mother and foetus. The use of the drug remdesivir is questionable due to the paucity of known data. Other drugs, such as lopinavir, ritonavir, chloroquine, corticosteroids and anticoagulants, are associated with small or unrepresentative safety concerns [4,16,21].

Table 2 lists, according to the FDA, the drugs selected in this literature review, distinguishing them according to the risk of teratogenicity and maternal, foetal and neonatal complications [2].

### Limitations

Despite the considerable increase in the number of studies currently being produced against SARS-CoV-2, most of these studies exclude pregnant women, so there is a great lack of knowledge in the approach to this type of patient.

SARS-CoV-2, although linked to SARS in 2003, has different characteristics, which has led to great uncertainty about how to deal with it.

The few studies published so far on treatments in pregnant women preclude generalisation as they are case series and rather insignificant samples of participants.

Moral source is one of the major limitations to the inclusion of pregnant women in studies, leading to the denial of the use of therapeutic agents against COVID-19 due to lack of knowledge and minimal data on safety in pregnancy.

## 5. Conclusions

Due to the paucity of studies on the treatment of pregnant women with COVID-19, the data, currently from case series and studies with rather small sample sizes, have to be interpreted with caution as they cannot be generalised to obtain reliable and safe clinical results for treating such patients who are highly vulnerable during pregnancy.

There is an urgent need for pregnant women not to be excluded from clinical trials to obtain a solid basis on which to offer optimal pharmacological treatments that do not entail teratogenic risk for the mother and foetus throughout pregnancy, including possible subsequent malformations in the newborn. However, the limited data currently available are proving to be quite useful, effective and encouraging in the treatment of this type of patient.

The nursing profession should carry out clinical trials based on the approach to the pharmacological treatment of COVID-19 in pregnant women, establishing adequate and quality care plans, based on the needs demanded by this type of patient.

## Figures and Tables

**Table 2 jcm-10-04896-t002:** FDA classification.

Drug	FDA Category
Anticoagulants	B
Corticosteroids	C, D
Convalescent or hyperimmune plasma	No data
Remdesivir	B
Chloroquine/hydroxychloroquine	C
Lopinavir/ritonavir	C
Azithromycin	B

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
