# Peer review of "Pharmacological Treatments against COVID-19 in Pregnant Women"

_jcm, 2021, doi:10.3390/jcm10214896_

Round 1

Reviewer 1 Report

Please modify the Result section: the division in chapters is not useful. You must describe the Results in a more homogenous way.

You must be aware that hydroxycloroquine is a common treatment for pregnant women with autoimmune diseases and that the use of hyperimmune plasma doesn't seem useful in COVID -19 patients.

Table 2 must be included in the Results

Author Response

Dear reviewers,
Thank you very much for your reviews of this article, which have improved and updated its content. We hope we have been able to respond to the issues you have raised.
Reviewer 1: We appreciate your feedback on the removal of the subsections. They have been removed and we have included the full name the first time each of the analysed drugs appears.
Regarding the results on hydroxychloroquine and hyperimmune plasma, we have included the clarifications you pointed out in the text, clarifying that hydroxychloroquine is used in patients with autoimmune diseases and that hyperimmune plasma does not prove to be an effective treatment in pregnant women against covid-19, at the moment. 
Table 2 has been placed before the discussion, in the place of the results, as you have indicated to us.
Once again, we thank you for your feedback on this article and would be pleased to answer any questions you may have.
With best regards.

Reviewer 2 Report

Thank you for the opportunity to review the manuscript Anna Arco Torres et al.
#Pharmacological treatments against COVID-19 in pregnant women.#
The manuscript
meets the conditions for a systematic review. The review report met the criteria
of the Preferred Re-porting Items for Systematic reviews and Meta-Analyses
[PRISMA] review protocol and was registered on the website:
http://www.crd.york.ac.uk/PROSPERO/ with the registration number CRD42021251036,
which increases its cognitive value. The problem of covid19 therapy in pregnant
patients is of great importance in the current pandemic situation and collecting
research on this subject allows you to learn about proven and modern approaches
in these cases.
In the systematic review by Anna Arco Torres et al., the 22 most
valuable works were found correctly and according to the rules.
Authors should
consider whether case reports are necessary in this work.
The manuscript is
consistent, readable, and the literature is very fresh.

Author Response

Dear reviewers,
Thank you very much for your reviews of this article, which have improved and updated its content. We hope we have been able to respond to the issues you have raised.
Reviewer 2: suggests that we should clarify whether the clinical cases reviewed are necessary, so we have included the comment in the text that they are necessary because they provide a detailed account of the symptoms, medical signs, diagnosis and therapeutic follow-up of pregnant women diagnosed with covid 19, which is the focus of this paper.
Once again, we thank you for your feedback on this article and would be pleased to answer any questions you may have.
With best regards.